# Involvement of M1-Activated Macrophages and Perforin/Granulysin Expressing Lymphocytes in IgA Vasculitis Nephritis

**DOI:** 10.3390/ijms25042253

**Published:** 2024-02-13

**Authors:** Gordana Laskarin, Emina Babarovic, Nastasia Kifer, Stela Bulimbasic, Mario Sestan, Martina Held, Marijan Frkovic, Alenka Gagro, Marijana Coric, Marija Jelusic

**Affiliations:** 1Department of Physiology, Immunology and Pathophysiology, Faculty of Medicine, University of Rijeka, 51000 Rijeka, Croatia; 2Hospital for Medical Rehabilitation of Hearth and Lung Diseases and Rheumatism “Thalassotherapia-Opatija”, 51410 Opatija, Croatia; 3Department of Pathology, Faculty of Medicine, University of Rijeka, 51000 Rijeka, Croatia; emina.babarovic@uniri.hr; 4Division of Rheumatology and Immunology, Department of Paediatrics, School of Medicine, University Hospital Centre Zagreb, University of Zagreb, 10000 Zagreb, Croatia; nastasia.ce@gmail.com (N.K.); mario.sestan@gmail.com (M.S.); martinaheld17129@gmail.com (M.H.); mfrkovic1@gmail.com (M.F.); marija.jelusic@mef.hr (M.J.); 5Department of Pathology and Cytology, School of Medicine, University Hospital Centre Zagreb, University of Zagreb, 10000 Zagreb, Croatia; sbulimba@kbc-zagreb.hr (S.B.); marijana.coric17@gmail.com (M.C.); 6Children’s Hospital Zagreb, Faculty of Medicine, University of Osijek, 31000 Osijek, Croatia; alenka.gagro@gmail.com

**Keywords:** granulysin, Henoch–Schönlein purpura nephritis, IgA vasculitis nephritis, macrophage polarisation, NK cells, perforin, T cells

## Abstract

We investigated the polarisation of CD68+ macrophages and perforin and granulysin distributions in kidney lymphocyte subsets of children with IgA vasculitis nephritis (IgAVN). Pro-inflammatory macrophage (M)1 (CD68/iNOS) or regulatory M2 (CD68/arginase-1) polarisation; spatial arrangement of macrophages and lymphocytes; and perforin and granulysin distribution in CD3+ and CD56+ cells were visulaised using double-labelled immunofluorescence. In contrast to the tubules, iNOS+ cells were more abundant than the arginase-1+ cells in the glomeruli. CD68+ macrophage numbers fluctuated in the glomeruli and were mostly labelled with iNOS. CD68+/arginase-1+ cells are abundant in the tubules. CD56+ cells, enclosed by CD68+ cells, were more abundant in the glomeruli than in the tubuli, and co-expressed NKp44. The glomerular and interstitial/intratubular CD56+ cells express perforin and granulysin, respectively. The CD3+ cells did not express perforin, while a minority expressed granulysin. Innate immunity, represented by M1 macrophages and CD56+ cells rich in perforin and granulysin, plays a pivotal role in the acute phase of IgAVN.

## 1. Introduction

Immunoglobulin A vasculitis (IgAV), previously known as Henoch–Schönlein purpura (HSP), is the most common form of systemic small-vessel vasculitis, with an annual incidence ranging from 3 to 55.9 cases per 100,000 children [1,2]. It is caused by galactose-deficient IgA1 [3,4], which is believed to be most frequently produced following upper respiratory tract infections, particularly those caused by beta-haemolytic *Streptococcus* [5]. Antiglycan antibodies recognise galactose-deficient IgA1 and form immune complexes with small molecular masses [3,4]. These complexes are deposited in the skin, joint structures, and/or the gut of patients with IgAV, correlating with increased concentrations of IgA and IgG in the serum as well as increased circulation of CD19+ B cells [6]. Approximately 30–50% of children with IgAV develop nephritis (IgAVN) [5]. Children with IgAVN have additional large molecular mass immune complexes in the circulation that are deposited in the glomerular mesangium and trigger subsequent immune reactions and kidney injury [5]. This coincides with a more significant decrease in IgM, C3, and C4 levels in children with IgAVN than in those with IgAV [6], indicating alternative complement activation by immune complexes [3]. The stimulation of macrophages with activated complement components can skew their polarisation toward a pro-inflammatory macrophage (M)1 phenotype [7] and monokine secretion, which significantly shapes cell-mediated immunity [8].

In children with IgAVN, peripheral blood natural killer (NK) and CD3+ T cell numbers, including CD4+ and CD8+ T cell subtypes, were decreased compared to those in controls, and were significantly lower than those in children with IgAV [6,9]. Contrarily, Fengyong et al. [10] claimed that peripheral blood CD8+ cell numbers increased in children with IgAVN compared to those in children with IgAV but displayed an inactive phenotype with reduced activation marker CD69 levels and decreased pro-inflammatory cytokine, tumour necrosis factor-alpha (TNF-α), and interferon-γ secretion. Similarly, peripheral blood NK cell cytotoxicity is attenuated, and circulating NK cells exhibit decreased expression of the activating natural cytotoxicity receptors NKp30 and NKp46 [11]. These findings suggest that cell-mediated immunity participates in the pathogenesis of IgAVN along with humoral immunity, depending on disease activity [9]. Peripheral blood NK cells and cytotoxic T lymphocytes are activated during the acute stage of the disease and decrease during the convalescent stage, as indicated by human leukocyte antigen (HLA)-DR expression [12].

At the gene level, messenger ribonucleic acid (mRNA) levels of cytotoxic mediators, including granulysin and granzyme B, which represent cell-mediated immunity, are significantly higher in the peripheral blood mononuclear cells of children with IgAVN than those in healthy controls [12]. However, limited data are available regarding local kidney cell-mediated immunity in children with IgAVN.

Recent findings through single immunohistochemical labelling revealed the recruitment of CD8+ lymphocytes, natural killer (NK) cells, and cells expressing apoptotic mediators such as granulysin and granzyme B in the glomeruli of children with IgAVN compared to healthy controls [12]. In human cells, granzyme B and mature, short 9 kDa forms of granulysin are densely packed with the pore-forming cytolytic molecule perforin within cytotoxic granules in one pole of activated effector cells [13]. They are released on demand after transmitting the activation signal to the cells in a calcium-dependent manner [13]. The released perforin monomers diffuse through the immune synaptic cleft to the target cell membrane and aggregate to form pores, allowing the influx of extracellular water into the cytoplasm and resulting in target cell swelling and necrosis [14]. Perforin pores allow for the rapid influx of apoptotic molecules such as granzyme B, Fas ligand, and the cytotoxic form of granulysin (9 kDa) from effector cells into target cells [14]. This enables the activation of apoptotic mechanisms in the cytoplasm and nucleus of target cells [12,14,15]. Owing to its complex mechanism of action, perforin serves as a biological marker for strong pro-inflammatory immune responses in various pathophysiological conditions, such as acute myocardial infarction [16], sepsis [17], and pregnancy loss [18]. Its expression in peripheral blood lymphocytes is accompanied by an increase in granulysin expression and granulysin-mediated cytotoxicity [19,20]. Increased granulysin mRNA expression has been observed in transplant biopsies during acute kidney rejection or infection [21]. Both perforin and granulysin are associated with M1 polarisation, which is typically characterised by inducible nitric oxide synthase (iNOS) production and the secretion of a set of pro-inflammatory monokines and chemokines that can promote tissue injury rather than tissue repair, as seen in M2 polarised arginase-1 positive macrophages [22].

To the best of our knowledge, there are no available data on the polarisation of CD68+ cells in the kidneys of children with IgAVN or their role in supporting perforin production by T and NK cells. In this study, we aimed to investigate the polarisation of CD68+ cells, examine the spatial tissue interaction between CD68+ and CD3+ or CD56+ cells, and explore the cytotoxic phenotypes of CD3+ and CD56+ cells (granulysin, perforin, and NKp44 expression) in the glomeruli and tubules of kidney biopsy specimens obtained from patients with IgAVN.

## 2. Results

### 2.1. Characteristics of the Children Recruited for the Study

The characteristics of the children enrolled in this study are shown in the Table 1.

### 2.2. Assessment of Glomerular CD68+ Cell Polarisation and Spatial Distribution with Respect to CD3+ and CD56+ Lymphocytes

To analyse the polarisation of CD68+ cells, we performed double labelling of CD68 and the M1 marker, iNOS, or the M2 marker, arginase-1, using immunofluorescence in paraffin-embedded kidney sections from biopsies of patients with IgAVN (Figure 1). The CD68 antigen (red fluorescence) was observed in the cytoplasm and on the cell surface of certain glomerular and tubular epithelial cells (Figure 1A,B, respectively). Most of these cells expressed the iNOS antigen (green fluorescence); hence, the co-expression of CD68 and iNOS after merging the microphotographs manifested yellow fluorescence (Figure 1A,B). In the combination of CD68/agrinase-1 labelling, CD68+ cells appeared as green fluorescence in the glomerulus (Figure 1C), interstitium, tubular epithelial cells, and even inside the tubules (Figure 1D). Arginase-1+ cells were not detected in the glomerulus in the shown sample (Figure 1C) but were present in the tubular epithelial cells (Figure 1D). Therefore, the co-expression of CD68 and arginase-1 presented as yellow fluorescence only in tubular epithelial cells after merging the CD68 and arginase-1 microphotographs (Figure 1D).

In the glomerulus rich in immune cells (Figure 2A), CD68+ cells with a peripheral glomerular distribution (red fluorescence,) surrounded the CD3+ cells (green fluorescence). The close contact between CD68+ cells (red fluorescence) and CD56+ cells (green fluorescence) is shown in Figure 2B.

### 2.3. Cytotoxic Phenotype of Kidney Infiltrating Lymphocytes

The phenotype was analysed according to cytotoxic NKp44 activating receptor expression in CD56+ cells (Figure 2C) and the expression of cytotoxic mediators perforin (Figure 3) and granulysin (Figure 4) in CD3+ and CD56+ lymphocyte subsets. Labelled CD56+ cells (red fluorescence) and NKp44+ cells (green fluorescence) were observed at the same locations in the glomeruli, suggesting that identical cells were labelled with both antibodies (Figure 2C). Their co-expression was confirmed by yellow fluorescence after merging the CD56+ and NKp44+ micrographs. Tubules with more damaged structures showed higher nonspecific antibody binding than tubules made of cells with regular, densely distributed nuclei (Figure 2C).

In sections double labelled with anti-CD56 and antiperforin antibodies, more perforin expressing CD56+ cells were observed in the glomeruli (G part, surrounded by a dashed curve) (Figure 3A) than in Figure 3B (red fluorescence), indicating variable infiltration in the glomeruli in the same patient with IgAVN. The number of CD56+ cells in the interstitium and between tubular epithelial cells was very low (T part, red fluorescence; Figure 3B). In contrast, CD3+ cells were less abundant in the glomerulus (G part), but infiltrated the interstitium more between the tubules (T part) (Figure 3C,D, respectively).

Moreover, perforin expression differed between CD56+ and CD3+ lymphocyte subsets. Perforin (green fluorescence) was expressed in the glomerular NK cells (G part, Figure 3B) and the interstitial NK cells (T part, Figure 3B). Overlapping perforin positive and CD56+ photomicrographs revealed that almost all CD56+ cells were perforin positive. Perforin was mostly found at the pole of CD56+ cells. In the shown sample, antiperforin and anti-CD56 antibodies remained sufficiently long on the tissue section during the labelling procedure and were retained in the glomerular vascular spaces (Figure 3A). Perforin-expressing CD3+ cells were absent in the glomerulus (G part, Figure 3C), in the interstitium and intraepithelially (Figure 3D). Tubuli of damaged structures were bound to anti-CD56 or anti-CD3 and antiperforin monoclonal antibodies, and yellow fluorescence appeared after merging the photomicrographs (Figure 3B–D).

Labelling of CD56 (red fluorescence) and granulysin (green fluorescence) revealed granulysin-positive CD56+ cells in the glomerulus (G part, Figure 4A), and tubule lumen (Figure 4B) as indicated by the yellow arrows. The CD56+ cells infiltrating the interstitium did not express granulysin (Figure 4B). Labelling of CD3 (red fluorescence) and granulysin (green fluorescence) revealed that a part of CD3+ lymphocytes co-expressed granulysin in the glomeruli (Figure 4C, part G) and could be found in the tubule lumen (Figure 4D), as indicated by the yellow arrows. Some glomerular (Figure 4C, part G), interstitial (Figure 4C, part T) and intraepithelial (Figure 4D) CD3+ cells did not express granulysin, as indicated by the white arrows in the overlapping photomicrographs. Damaged tubules with rarer and irregularly arranged nuclei fluoresced red and green in samples double labelled with antigranulysin and anti-CD56 (Figure 4B) or anti-CD3 (Figure 4D) antibodies and appeared as yellow fluorescence after overlapping photomicrographs. Isotype controls are shown in the Appendix A for all labelling described above. Table 2 summarises the main results of the study.

## 3. Discussion

We demonstrated the presence of the cytolytic mediator perforin in the kidneys of children with IgAVN, although perforin has been previously investigated in various human kidney diseases and experimental models. In rat glomeruli, after the induction of mesangial proliferative glomerulonephritis, perforin mRNA expression increases along with T and NK cell infiltration [23]. In adults with IgA nephritis, increased perforin mRNA levels in peripheral blood T cells positively correlated with glomerular histopathology grade and proteinuria [24]. Subsequently, perforin was identified as a serum and urine biomarker of kidney allograft dysfunction and rejection owing to its expression in cytotoxic T cells [25]. However, our results demonstrated that in children with IgAVN, perforin was predominantly distributed in glomerular CD56+ lymphocytes, which mediate early innate immune responses [26]. This is consistent with the finding that kidney NK cells increase the content of perforin cytotoxic protein in mice with active nephritis in a model of systemic lupus erythematosus (SLE), contributing to kidney damage and disease outcome [27]. Similarly, in patients with active SLE nephritis, mature peripheral blood CD56(dim) NK cells expressing perforin and CXC receptor 3 leave the circulation and are recruited to the glomeruli, reflecting the movement of NK cells from the blood to the target organ [28]. The glomerular autoimmune microenvironment promotes further maturation of NK cells together with antigen-presenting cells, as both express receptors for the Fcγ chain [29]. The close spatial relationship between a few glomerular-specific activated pro-inflammatory M1 macrophages and one NK cell in children with IgAVN may significantly influence the phenotype and function of NK cells through multiple interactions, including the development of a cytotoxic phenotype [8]. We demonstrated the expression of the cytotoxic NKp44 receptor in glomerular CD56+ cells, suggesting that they are prone to cytotoxic activity after direct contact with target glomerular cells, in line with the fact that the depletion of NK cells in a mouse experimental model prevents the occurrence of glomerulonephritis [29]. Cytotoxic CD56+ cells of the NKp44+ phenotype reach the tubular lumen after leaving the damaged glomeruli through cytotoxic mechanisms, as demonstrated by perforin-expressing CD56+ cells within the tubal lumen. NK cells in the tubules may be responsible for perforin-mediated necrosis of tubular epithelial cells, as damaged tubular epithelial cells were labelled with an antiperforin antibody in our experiments, and perforin-mediated acute tubular necrosis is a mechanism of tubular damage in IgAVN [26]. A similar observation was made in the kidney during acute postinfectious glomerulonephritis, focal necrotising (pauci-immune) glomerulonephritis [30], acute kidney rejection [31], or infection [32], indicating strong cell-mediated immunity in IgAVN. Additionally, we observed granulysin expression in glomerular and intratubular CD56+ cells but not in interstitial CD56+ cells, suggesting an apoptotic mechanism of tubular damage, primarily by CD56+ cells of glomerular origin, as observed in the acute rejection of kidney transplants resistant to glucocorticoid treatment [21]. This is visualised by granulysin labelling of damaged tubular epithelial cells, aligned with the presence of granulysin in the kidney tissue of IgAVN [12].

In the glomeruli of children with IgAVN, CD56+ lymphocytes can be considered NK cells, even though CD56 can be expressed on T cell subsets [33]. This is supported by the significantly lower representation of CD3+ T lymphocytes in the glomeruli of children with IgAVN than that of CD56+ cells [33]. Furthermore, CD3+ lymphocytes did not express perforin in either the glomeruli or the kidney interstitium, indicating a partially inactive phenotype of T cells in the early and clinically acute phases of the disease at biopsy. This aligns with the inactive phenotype of peripheral blood CD3+, CD4+, and CD8+ lymphocytes, which is characterised by a decrease in CD69 expression and cytokine secretion, concurrent with an increase in serum creatinine and proteinuria in children with IgAV, particularly in those with IgAVN [10], given that T lymphocytes enter the kidney tissue from the circulation [12] in a proteinuria-dependent manner [34]. Some glomerular CD3+ lymphocytes expressed granulysin and CD3+ lymphocytes in the tubular lumen, which likely originated from damaged glomeruli, confirming the initiation of CD3+ cell activation.

CD68+ cells establish closer contact with only some glomerular CD3+ lymphocytes, which become granulysin- rather than perforin-positive cells and possibly mark only the beginning of the activation of acquired immunity. This finding aligns with the observation that granulysin appears only in activated peripheral blood T cells [35]. We speculated that a long non-cytotoxic 15 kD form of granulysin could be expressed in glomerular T lymphocytes and stored in perforin-negative granules instead of granulysin (9 kDa) with cytotoxic properties, which is stored only in perforin-positive polarised granules of lymphocytes [13,36]. However, we were unable to confirm our hypothesis because of the use of a commercial RC8 antigranulysin antibody, which recognises the common epitope of both forms of granulysin [37]. The 15-kDa granulysin form is not cytotoxic and acts as an intensifier of inflammation by increasing the chemotaxis of monocytes, memory T lymphocytes, NK cells, monocytes, and mature dendritic cells from the circulation into the tissue, even at nanomolar concentrations [37]. This was confirmed by the positive correlation observed between the frequency of T lymphocytes and macrophages in the glomeruli of patients with IgAVN [33]. The long 15-kDa granulysin differentiates human monocytes into macrophages [38] and stimulates the production of monokines and chemokines [36] at the beginning of the cell-mediated immune response [36]. Owing to the superiority of macrophages over certain lymphoid and non-haematopoietic cells that can be labelled with anti-CD68 antibodies, CD68 is widely considered a marker of tissue macrophages [39]. The shape and average number of CD68+ macrophages per glomerulus in our children with IgAVN varied and may be dependent on disease severity [40]. Macrophages are plastic cells that respond to numerous stimuli from the immediate microenvironment that change their transcriptional profiles and phenotypes, leading to a fine line between inflammation and tissue remodelling [22]. Accordingly, the dominance of CD68+ macrophages expressing iNOS over arginase-1 in the glomeruli indicated strong M1 orientation [22].

## 4. Materials and Methods

### 4.1. Patients

Children with IgAV and significant proteinuria diagnosed according to the criteria established by the European League against Rheumatism, Paediatric Rheumatology International Trials Organization, and Paediatric Rheumatology European Society [41,42] were enrolled in this study. Percutaneous renal biopsy was performed as a routine clinical diagnostic procedure for clinically active IgAVN in patients with 24-h proteinuria > 1.0 g/dU, persistent 24-h proteinuria > 0.5 g/dU for at least three months, nephritic and nephrotic syndrome and impaired eGFR [43,44] and paraffin-embedded kidney biopsy specimens were archived at the Department of Pathology and Cytology of the University Hospital Centre Zagreb. For the purposes of this research, paraffin blocks were retrogradely selected from the children (N 30) who suffered from IgAVN in the period from year 2015. None of the enrolled children exhibited any other systemic diseases, as indicated in their medical records. The exclusion criteria were other autoimmune illnesses, immune deficiency, bone marrow diseases, lymphatic system disorders, diabetes (glucose level > 11 mmol/L), chronic liver disease, heart disease, organ injury, previous blood transfusions, and malignant diseases.

### 4.2. Section Processing

Fresh kidney tissue samples were obtained via percutaneous renal biopsy [45] and immediately embedded in paraffin using a previously described method [46]. Subsequently, before labelling, the paraffin block was cut using Cryocut Leica, SM2010R (Leica Biosystems, Nussloch, Germany) on 3 ug slides. Kidney tissue sections were mounted on HistoBond 75 × 25 × 1-mm microscope glass (Marienfeld, Lauda-Koningshofen, Germany), placed in plastic racks, and immersed in plastic cuvettes (Marienfeld) containing Tissue Clear (xylene-based paraffin remover; Sakura Fintek Europe, Zoeterwoude, Netherlands) for 5 min at room temperature in a digester. The hydration process involved sequential immersion of the tissue sections in 100% ethanol (Kemika, Zagreb, Croatia) for 3×5 min, 96% ethanol for 2 × 5 min, 75% ethanol for 5 min, and PBS [Engl. Phosphate-buffered saline, 0.05 M containing 0.3 M NaCl; pH 7.4 (Kemika, Zagreb, Croatia)] for 5 min. All the steps were conducted at room temperature using a digester. Detection of epitopes of interest (CD68, iNOS, arginase-1, CD3, CD56, perforin, granulysin, and NKp44) for tissue labelling by immunofluorescence was performed by boiling tissue sections in 10 mM citrate buffer (pH 6.0) in a microwave oven (3 × 70 W for 5 min) while replacing the spent distilled water by boiling. After cooling to room temperature for approximately 1 h, the tissue sections were washed with TBS (PBS 0.05% Tween20, Sigma-Aldrich Chemie, St. Louis, MO, USA). Surrounding the tissues with “DAKO-Pen” (DAKO, Glostrup, Germany) facilitated the application of chemicals and antibodies to horizontally laid slides in a humid and dark chamber.

### 4.3. Double Immunofluorescence Labelling

Immunofluorescence labelling was performed on paraffin-embedded kidney sections from patients with IgAV (3 μm) after the section processing, utilising a previously described method [47]. To minimise nonspecific binding, the slides were incubated with 1% bovine serum albumin (BSA; Sigma-Aldrich) in PBS for 1 h at room temperature. The tissue sections were subsequently incubated with a combination of mouse anti-CD68 mAb and rabbit anti-iNOS, anti-NCAM-1 (CD56), or anti-CD3 antibodies. Rabbit anti-CD68 mAb was combined with mouse anti-arginase-1 antibody. Rabbit anti-CD3 antibody was combined with mouse antiperforin and antigranulysin antibodies. The rabbit anti-NCAM (CD56) was incubated with mouse antibodies against perforin, granulysin, or anti-NKp44. Appropriate isotype controls (mouse IgG1, IgG2b, or IgG3) were used in combination with rabbit IgG. Secondary antibody, Alexa Fluor 594 donkey antimouse (red fluorescence) and Alexa Fluor 488 donkey antirabbit (green fluorescence), were used to visualise CD68/iNOS, arginase-1/CD68, CD68/CD3, and CD68/CD56 labels. Secondary antibody, Alexa Fluor 594 goat anti rabbit (red fluorescence) and Alexa Fluor 488 goat antimouse (green fluorescence) were used to visualise CD56/NKp44, CD56/perforin, CD56/granulysin, CD3/perforin, and CD3/granulysin. Nuclei were visualised using 4,6-diamidino-2-phenylindole (DAPI; Sigma-Aldrich) diluted 1:1000 in PBS for 5 min at room temperature. The slides were then washed three times with PBS and mounted using Mowiol mounting medium (Sigma-Aldrich). All the antibodies used in this study are listed in Appendix A.

### 4.4. Analysis of Immunofluorescent Labelling

An average of 7 glomeruli (range 5–14 glomeruli) in each sample were visualised, and the representative glomeruli were photographed. The tubules and interstitium were analysed separately in the 10 locations with the highest density of positive cells (“hotspots”) and the representative fields were photographed. Fluorescent images were acquired with an Olympus BX51 fluorescent microscope using an Olympus DP71 camera (Olympus) and Olympus UPlan objective lens (Apo100×/1.35 Oil Iris; ×1000). CellA imaging software (version 3.0; Olympus) was used to acquire and overlay images displaying red and green fluorescence in the double-labelled tissue sections. In one tissue section double-labelled with antibodies, one spot (field of view at 1000 magnification) was photographed in three ways: (1) using Red Excitation (650 nm) to maximally present deep red emission (690 nm); (2) using Blue Excitation (450 nm) to maximally present Green Emission (550 nm); (3) using Ultraviolet Excitation (350 nm) to maximise the blue emission (450 nm) of the nuclei stained with DAPI. Overlapping microphotographs (1) and (3) shows cells with nuclei and red fluorescence, whereas the overlapping microphotographs (2) and (3) cells, with nuclei and green fluorescence. By overlapping the microphotographs, (1), (2), and (3) co-expression of the antigens of interest in the cells with the nucleus was visualised. High-resolution images were acquired using Fiji program [48].

## 5. Conclusions

The expression of perforin and granulysin in NK cells originating from the glomeruli, along with the co-expression of NKp44 in children with IgAVN, proves the strong cytotoxic potential of innate immune cells during the early acute phase of the disease in a pro-inflammatory tissue environment, supported by M1 CD68+ macrophages. Analysis of cell-mediated immunity in a kidney biopsy sample of early active disease could help select treatment options [43,44], as immunosuppressive remedies target lymphocytes and are effective against IgAVN [33,47]. However, further studies are required to investigate the activation mode of NK cells in the glomerulus, intracellular distribution of other cytotoxic mediators, relationship between activating and inhibiting receptors, and cytotoxic signalling pathways in NK cells infiltrating the kidney in children with IgAVN.

## Figures and Tables

**Figure 1 ijms-25-02253-f001:**
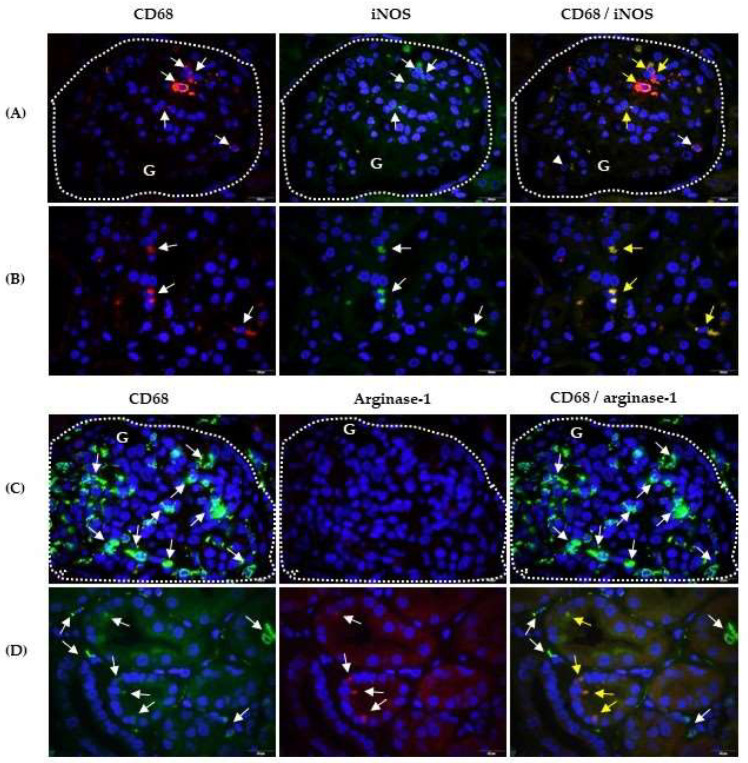
CD68/iNOS and CD68/arginase-1 co-expression in the kidneys of patients with IgAV nephritis. Immunofluorescence assay shows double labelling with anti-CD68 mouse IgG1 (red) and anti-iNOS rabbit IgG (green) in the glomeruli-G (**A**) and in the tubules (**B**) or double labelling with anti-CD68 rabbit IgG (green) and anti-arginase-1 mouse IgG3 (red) in the glomeruli (**C**) and in the tubules (**D**). White arrows indicate cells with red or green fluorescence. The co-expression of CD68 and iNOS or arginase-1 is visualised in yellow in the last column (yellow arrows). The nuclei are stained blue with 40,6-diamidino-2-phenylindole. White dashed curves mark the boundary of the glomeruli (G). One of eight experiments conducted in each group is shown. Magnification was achieved by the Olympus UPlan objective lens Apo100×/1.35 Oil Iris (×1000).

**Figure 2 ijms-25-02253-f002:**
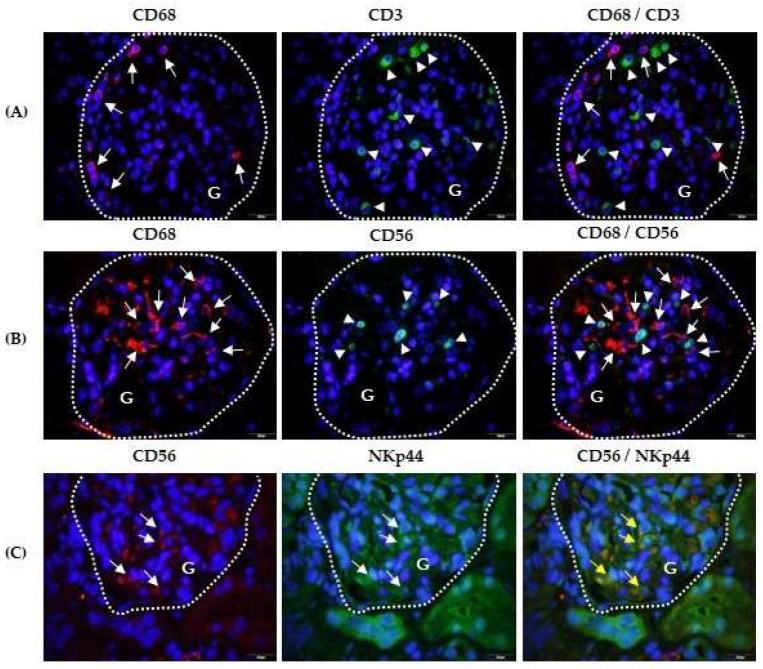
Spatial relationships among CD68, CD3, and CD56 positive cells, and CD56 and NKp44 co-expression in glomeruli (G) of patients with IgAV nephritis. Immunofluorescence shows double-labelled anti-CD68 mouse IgG1(red, white arrows) and anti-CD3 rabbit IgG (green, white arrowheads) (**A**), anti-CD68 mouse IgG1(red, white arrows) and anti-CD56 rabbit IgG (green, white arrowheads) (**B**), and anti-CD56 rabbit polyclonal (red, white arrows) and anti-NKp44 mouse IgG2a (green, white arrows) (**C**). The co-expression of CD56 and NKp44 is shown in yellow in the last column (yellow arrows). Nuclei are stained blue with 40,6-diamidino-2-phenylindole. White dashed curves mark the boundary of the glomeruli (G). One of the eight experiments conducted in each group is presented herein. Magnification was achieved using an Olympus Uplan objective lens Apo100×/1.35 Oil Iris (×1000).

**Figure 3 ijms-25-02253-f003:**
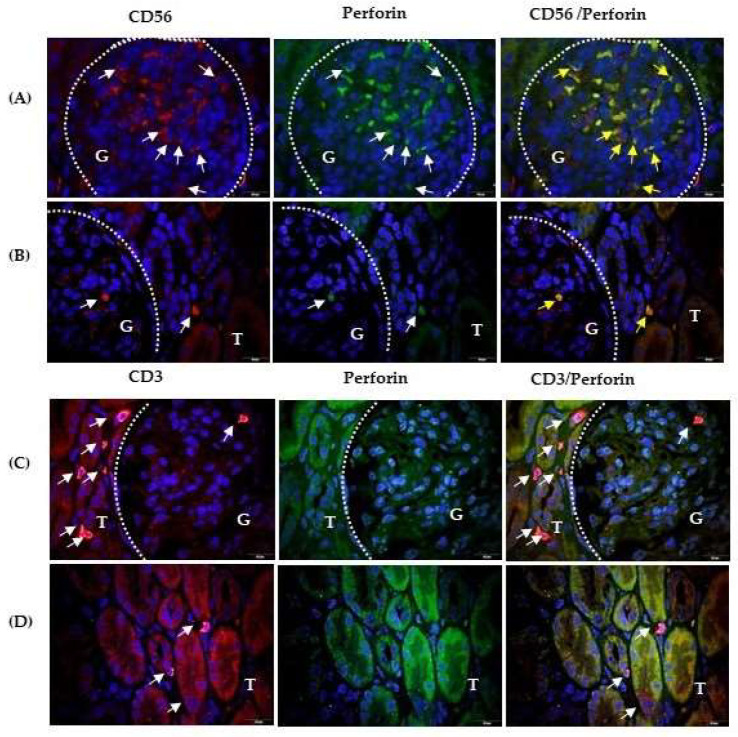
CD56/perforin and CD3/perforin co-expression in kidney of the patients with IgAV nephritis. Immunofluorescence revealed double labelling with anti-CD56 rabbit IgG (red) and antiperforin mouse IgG2b (green) in the glomeruli (**A**) and tubules (**B**), or double labelling with anti-CD3 rabbit IgG and antiperforin mouse IgG2b in the glomeruli (**C**) and tubules (**D**). White arrows indicate cells with red or green fluorescence. Co-expression of CD56 and perforin is shown in yellow in the last column (yellow arrows). Nuclei were stained blue with 40,6-diamidino-2-phenylindole. White dashed curves mark the boundary between the glomeruli (G) and tubules (T). One of the eight experiments is shown and described. Magnification was achieved using an Olympus UPlan objective lens Apo100×/1.35 Oil Iris (×1000).

**Figure 4 ijms-25-02253-f004:**
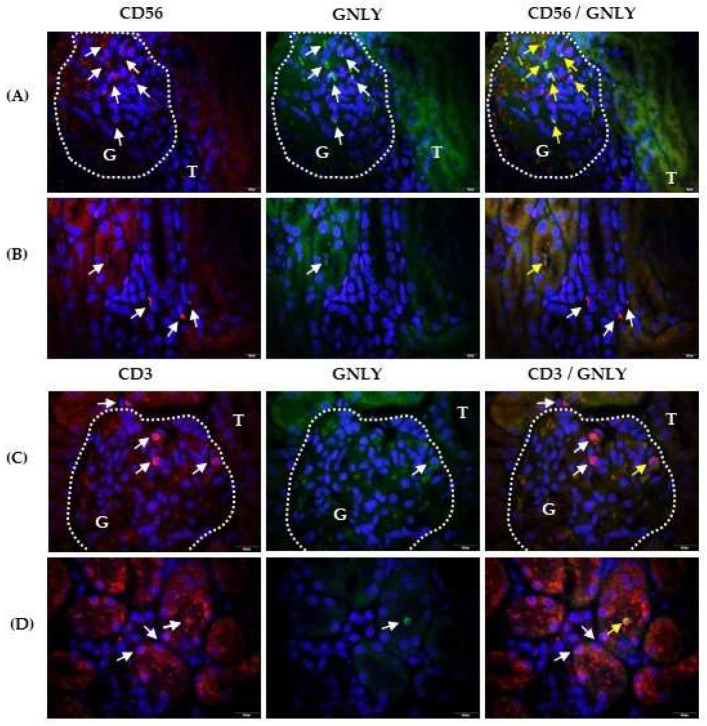
CD56/granulysin and CD3/granulysin co-expression in the kidneys of patients with IgAV nephritis. Immunofluorescence shows double labelling with anti-CD56 rabbit IgG (red) and antigranulysin mouse IgG1 (green) in the glomeruli (**A**) and the tubules (**B**), or double labelling with anti-CD3 rabbit IgG and antigranulysin mouse IgG1 in the glomeruli (**C**) and tubules (**D**). White arrows indicate cells with red or green fluorescence. Co-expression of CD56 and granulysin is shown in yellow in the last column (yellow arrows). White dashed curves mark the boundary between the glomeruli (G) and tubules (T). Nuclei were stained blue with 40,6-diamidino-2-phenylindole. One of the eight experiments conducted in each group is described above. Magnification was achieved using an Olympus UPlan objective lens Apo100×/1.35 Oil Iris (×1000).

**Table 1 ijms-25-02253-t001:** Demographic, laboratory, and clinical characteristics of patients recruited for the study.

Demographic Characteristics	
Female (%)	47
Age at diagnosis, years (median, q1- q3)	11.9 (7.7–13)
Laboratory parameters	
Creatinine, µmol/L (median, q1- q3)	52 (39–74)
Urea, mmol/L (median, q1- q3)	4.5 (3.5–5.2)
Haematuria (%)	93
24-h proteinuria, g/dU (median, q1- q3)	0.92 (0.33–1.97)
eGFR, ml/min/1.73 m^2^ (median, q1- q3)	114.1 (78.4–133.3)
ESR, mm/h (median, q1- q3)	17 (12–22)
C3 (median, q1- q3)	1.26 (1.132–1.332)
C4 (median, q1- q3)	0.24 (0.187–0.292)
Therapy	
NSAIDs (%)	39
Glucocorticoids (%)	96
Immunosuppressives (%)	43
Biologic therapy (%)	0
Antihypertensives (%)	65
Outcome	
normal physical examination, normal urinary and renal function (%)	36
normal physical examination, with microscopic haematuria and/or proteinuria < 1 g/day (or < 40 mg/h/m^2^), eGFR > 60 mL/min/1.73 m^2^ (%)	60
proteinuria > 1 g/day (or < 40 mg/h/m^2^) and/or hypertension, and eGFR > 60 mL/min/1.73 m^2^ (%)	4
eGFR < 60 mL/min/1.73 m^2^ or ESRD requiring dialysis and/or renal transplantation or death (%)	0

C, Complement component; eGFR, estimated Glomerular Filtration Rate; ESR, Erythricyte Sedimentation Rate; ESRD, End-Stage Renal Disease; NSAIDs, Nonsteroidal Anti-Inflammatory Drugs.

**Table 2 ijms-25-02253-t002:** The main results of the study.

	Results
1.	CD68+ macrophage numbers fluctuated in the glomeruli and were mostly labelled with iNOs in patients with IgAVN.
2.	CD68+/arginase-1+ cells were more frequently found in the tubules than in glomeruli of patients with IgAVN.
3.	CD56+ cells, enclosed by CD68+ cells, were more frequently found in the glomeruli and co-expressed NKp44, whereas the arrangement of CD68+ and CD3+ cells in the glomeruli was less close.
4.	The glomerular and intratubular CD56+ cells expressed perforin and granulysin, whereas glomerular and intratubular CD3+ cells expressed granulysin in a patient with IgAVN.
5.	CD56+ and CD3+ cells in the kidney interstitium showed neither perforin nor granulysin expression in patients with IgAVN.
6.	Epithelial cells of damaged and irregular tubules bound more antiperforin and antigranulysin mAbs.

CD, Cluster of Differentiation; Ig, Immunoglobulin; IgAVN, Immunoglobulin A vasculitis nephritis; iNOS, inducible Nitric Oxide Synthase; mAbs, monoclonal antibodies; NK, Natural Killer; NKp44, Natural Killer protein 44.

## Data Availability

Data available in a publicly accessible repository.

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
