# Peer review of "Involvement of M1-Activated Macrophages and Perforin/Granulysin Expressing Lymphocytes in IgA Vasculitis Nephritis"

_ijms, 2024, doi:10.3390/ijms25042253_

Round 1

Reviewer 1 Report

Comments and Suggestions for Authors

Laskarin et. al. investigate the location, number and expression of CD3 T cells, NK cells, and macrophages and activation status based on perforin, granulysin, and iNOS/arg1 expression, respectively, in kidneys of children with IgAVN. Little is known about the renal distribution of these subsets and the roles these molecules play in polarizing the local immune response.

This is a well-designed study for histologic analyses, but it has some drawbacks.

Lines 100-105 seem to imply that they are investigating the role perforin expression (in CD3T or NK cells?) plays in supporting polarization of antigen-presenting cells (APC) in the IgAVN kidney.  It is uncertain how IHC can address this.  And conversely, it seems more logical that APC would support production of perforin in T and NK cells.

Fig 1. Legend and manuscript text state that brown color indicates specific antigen staining, but brown color is not detected in the IHC images so it is uncertain what can be concluded here. It is also not clear what the arrows are pointing to.

Table 1. If this is based on figure 1 histology, it is unclear how the final numbers were determined for the presence of specific cells.

Fig 2. It is odd that CD68 and iNOS, or Arg, are expressed in almost exactly the same locations with the same pattern, especially in rows B&D.  They are almost completely co-localized. This seems more artefactual. The text says the arrows point to double staining (lines 160-161, 165), but the figure legend says they point to single stained cells. It is very uncertain what they are actually pointing to.

Figure 3. Why is the CD68 staining pattern so different between A and B? A seems quite cell-associated, but B almost looks like stromal tissue staining.

Figure 4. Surface staining of CD56 does not look uniform per cell, nor does it always look cell-associated, and perforin follows almost exactly the same physical pattern of staining as CD56 which is odd.

Figure 5. It seems that there is high background staining with granulysin making interpretation difficult.

Overall, the staining is just not clear enough to make firm conclusions as to what is being shown. And the arrows are confusing as to what exactly they are pointing out. Some cellular staining is convincing, but much is not and some appears it could be artefactual.

The study goals are reasonable and definitive results would be very useful to the field, but the staining must be improved so that results are conclusive.

Author Response

Please find answers to your comments below.

Laskarin et. al. investigate the location, number and expression of CD3 T cells, NK cells, and macrophages and activation status based on perforin, granulysin, and iNOS/arg1 expression, respectively, in kidneys of children with IgAVN. Little is known about the renal distribution of these subsets and the roles these molecules play in polarizing the local immune response.

This is a well-designed study for histologic analyses, but it has some drawbacks.

Lines 100-105 seem to imply that they are investigating the role perforin expression (in CD3T or NK cells?) plays in supporting polarization of antigen-presenting cells (APC) in the IgAVN kidney. It is uncertain how IHC can address this. And conversely, it seems more logical that APC would support production of perforin in T and NK cells.

You are right. We wrote what we did not mean to say. The sentence „To the best of our knowledge, there are no available data on perforin occurrence and its role in supporting the polarization of antigen-presenting cells in the kidneys of children with IgAVN.“ is corrected in the new version of the manuscript as follows: „To the best of our knowledge, there are no available data on the polarization of CD68+ cells in the kidneys of children with IgAVN and their role in supporting production of perforin in T and NK cells.“ The change is emphasized in the blue color font in the new version of the manuscript.

Fig 1. Legend and manuscript text state that brown color indicates specific antigen staining, but brown color is not detected in the IHC images so it is uncertain what can be concluded here. It is also not clear what the arrows are pointing to.

We performed labeling about a year ago and the sections were embedded in Acquatex mounting medium, as it was written in the manuscript. We counted the positive cells then, but we did not take photos at the same time. Aquatex cannot permanently preserve the brown color at the site of specific antigen binding, so the color changed in the most previously cells.

We also feel that cells of the CD68+, CD56+, CD3+, iNOS+, and arginase-1+ phenotypes are better represented in the remainder of the new version of the manuscript using double immunofluorescence, and therefore it is not necessary to show the immunohistochemistry results. Therefore, the authors decided and agreed not to include the results about the frequency of leukocyte cell subsets (old Figure 1 and old Table 1) in the new version of the manuscript. If we omit Figure 1 and Table 1, the introduction the aim of the study, the line of discussion and the conclusions remain unchanged. We adapted the manuscript accordingly and removed all data related to immunohistochemistry from the new version of the manuscript (Abstract, Results and Materials and Methods).

Table 1. If this is based on figure 1 histology, it is unclear how the final numbers were determined for the presence of specific cells.

The authors agreed not to include the results about the frequency of leukocyte cell subsets (Table 1) in the new version of the manuscript, as mentioned above.

Fig 2. It is odd that CD68 and iNOS, or Arg, are expressed in almost exactly the same locations with the same pattern, especially in rows B&D. They are almost completely co-localized. This seems more artefactual. The text says the arrows point to double staining (lines 160-161, 165), but the figure legend says they point to single stained cells. It is very uncertain what they are actually pointing to.

We performed double immunofluorescence in order to investigate the colocalization of CD68 and iNOS or the colocalization of CD68 and arginase-1 in the glomeruli and in the tubulointerstitium, as described in the text (line 160 -161, 165). Images of the old Figure 2 (new Figure 1) display red and green fluorescence in double-labelled tissue sections. In the legend of the old Figure 2 it is also written correctly: "Immunofluorescence shows double labeling with anti-CD68 mouse IgG1 (red) and anti-iNOS rabbit IgG (green) in the glomeruli-G (A) and in the tubules (B) or double labeling with anti-CD68 rabbit IgG (green) and anti-arginase-1 mouse IgG3 (red) in the glomeruli (C) and in the tubules (D).“

So, in one tissue section, which is double labeled with two different antibodies and the nucleus is stained with DAPI, one spot (field of view at magnification ´1000) is photographed in three manners:

1.) using Red Excitation (650 nanometers) light color to maximally present Deep Red Emission (690 nanometers);

2.) using Blue Excitation (450 nanometers) light color to maximally present Green Emission (550 nanometers);

3.) using Ultraviolet Excitation (350 nanometers) ligh color to maximally present the Blue Emission (450 nanometers) of nuclei marked with DAPI.

Overlapping microphotographs 1.) and 3.) showed the cells with nuclei and red fluorescence, and overlapping microphotographs 2.) and 3.) showed the cells with nuclei and green fluorescence. For the description of these photomicrographs, the sentence: "Single fluorescence is indicated by the arrows" is not written correctly, as You noticed. We wanted to say: „White arrows indicate cells with red or green fluorescence." It is writen in the new version of the manuscript with changes hightlighted in the blue font color (legend of the new Figure 1).

By overlapping microphotographs 1.), 2.) and 3.) the co-expression of CD68 and iNOS or arginase-1 in the cells with blue nucleus is visualized in yellow in the last column and marked using yellow arrows.

So it is not an artefact, but really evidence of co-expression of CD68 and iNOS or arginase-1 in the same cells. We briefly described the method under the title "Analysis of immunofluorescent labeling". Please find that text in the blue font in the new version of the manuscript with changes highlighted.

We are also here showing you a photomicrograph (magnification x400) to give you a better idea of the labelling pattern of CD68 and iNOS.

The example of anti-CD68 mouse IgG1 (red) and anti-iNOS rabbit IgG (green) labelling in kidney specimen of patient with IgAVN, magnification x 400

Figure 3. Why is the CD68 staining pattern so different between A and B? A seems quite cell-associated, but B almost looks like stromal tissue staining.

CD68 expressing cells are considered tissue macrophages, although CD68 can be expressed on other cells. They sense very well the environment in which they are located. Two different glomeruli are shown in the Figure 3A and 3B, so we assume that the macrophages are not at the same level of activation, because the CD68+ cells in Figure 3B are more branched with more extensions. We discussed it as follows : „Owing to the superiority of macrophages over certain lymphoid and non-haematopoietic cells that can be labelled with anti-CD68 antibodies, CD68 is widely considered a marker of tissue macrophages [39]. The shape and average number of CD68+ macrophages per glomerulus in our children with IgAVN varied and may be dependent on disease severity [40]. Macrophages are plastic cells that respond to numerous stimuli from the immediate microenvironment that change their transcriptional profiles and phenotypes, leading to a fine line between inflammation and tissue remodelling [22].“.

Figure 4. Surface staining of CD56 does not look uniform per cell, nor does it always look cell-associated, and perforin follows almost exactly the same physical pattern of staining as CD56 which is odd.

Perforin is constitutively expressed in almost all CD56+ cell cytoplasms and raises under inflammatory stimulation. It is ussually stored in one pole of the cells. We wrote in the Introduction: „In human cells, granzyme B and mature, short 9 kDa forms of granulysin are densely packed with the pore-forming cytolytic molecule perforin within cytotoxic granules in one pole of activated effector cells [13].“ We also inicated better the perforin expressing CD56+ cells in the new Figure 3 (old Figure 4) by arrows. In the sample shown on the new Figure 3A anti-perforin and anti-CD56 antibodies remained sufficiently long on the tissue section during the labeling procedure and were retained in the glomerular vascular spaces, suggesting that the positivity is not exclusively associated with cells (new Figure 3A). Therefore we added the sentences in the „Results“ as follows: „Perforin was mostly found at the pole of CD56+ cells. In the shown sample, anti-perforin and anti-CD56 antibodies remained sufficiently long on the tissue section during the labeling procedure and were retained in the glomerular vascular spaces (Figure 3A).“ Please find it in the new version of the manuscript with the changes highlighted.

Figure 5. It seems that there is high background staining with granulysin making interpretation difficult.

Tubules without regular structure and damaged epithelial cells were granulysin positive, because epithelial cells were probably damaged by cytotoxic mechanisms mediated by intratubular lymphocytes. We have found granulysin negative cells in the interstitium and glomerulus on the same photomicrograph, proving the specificity of the labeling.

Overall, the staining is just not clear enough to make firm conclusions as to what is being shown. And the arrows are confusing as to what exactly they are pointing out. Some cellular staining is convincing, but much is not and some appears it could be artefactual.

We increased DPI of the microphotographs to 1800 and automatically replaced the existing images before Your revision. It is likely that during this process the arrows shifted. Even the arrows ware missing in the old Figure 2A (upper left photo). We paid particular attention to the arrows in the Figures in the new version of the manuscript. Yellow arrows have been introduced instead of arrowheads to better point out the cells.

The study goals are reasonable and definitive results would be very useful to the field, but the staining must be improved so that results are conclusive.

We hope that the manuscript quality is improved with the corrections.

Thank you for teh critical reading the manuscript, and valuable suggestions.

Reviewer 2 Report

Comments and Suggestions for Authors

In the present paper, the authors investigated the presence and phenotype of CD68+ macrophages, as well as perforin and granulysin distributions, in kidney lymphocyte subsets of children with IgA vasculitis nephritis (IgAVN). To this aim, they have used both immunohistochemistry and immunofluorescence. They found that iNOS+ cells were predominant in the glomeruli while arginase-1+ cells were predominant in the glomeruli. CD68+ macrophage numbers in the glomeruli were mostly iNOS+. CD68+/arginase-1+ cells were abundant in the tubules. CD56+ cells were more frequent in the glomeruli than in the tubuli and co-expressed NKp44. The glomerular and intratubular CD56+ cells expressed perforin and granulysin, respectively. CD3+ cells were more abundant in the interstitium than in the glomeruli, with a minority expressing granulysin. The authors conclude that M1 macrophages and CD56+ cells rich in perforin and granulysin, plays a pivotal role in acute IgAVN phase.

Issues to address:

-The M&M sections must be improved. The authors need to indicate the demographic and clinical characteristics of all the patients recruited for the study (age, sex, drug treatment, blood biochemical data, prognosis…).

-The study lacks comparison groups: groups of patients with nephritis of different etiology should be included. Or at least, the authors should discuss the differences between the data presented in the current paper and similar studies performed on different nephritic patients

-Figure 1. The positivity for the antibody staining is not clear at all. No brown color is visible on the figure. Please advise.

-Figure 6 and table 2 can be provided as supplementary material

Comments on the Quality of English Language

Minor editing requred

Author Response

Please find answers to your comments below.

Issues to address:

-The M&M sections must be improved. The authors need to indicate the demographic and clinical characteristics of all the patients recruited for the study (age, sex, drug treatment, blood biochemical data, prognosis…).

We added the new Table 1 with the demographic, laboratory and clinical characteristics of all the children recruited in the study.

-The study lacks comparison groups: groups of patients with nephritis of different etiology should be included. Or at least, the authors should discuss the differences between the data presented in the current paper and similar studies performed on different nephritic patients.

Unfortunately we did not include patients with nephritis of different etiology in this investigation. In the discussion of the new  version of the manuscript we compared our results with the data of other groups investigating other types of nephritis in humans and in experimental models. Please find them emphasized in the blue color font in the new version of the manuscript (Discussion and References).

 -Figure 1. The positivity for the antibody staining is not clear at all. No brown color is visible on the figure. Please advise.

We performed labeling about a year ago and the sections were embedded in Acquatex mounting medium, as it was written in the manuscript. We counted the positive cells then, but we did not take photos at the same time. Aquatex cannot permanently preserve the brown color at the site of specific antigen binding, so the color changed in the most previously cells. 

We also feel that cells of the CD68+, CD56+, CD3+, iNOS+, and arginase-1+ phenotypes are better represented in the remainder of the new version of the manuscript using double immunofluorescence, and therefore it is not necessary to show the immunohistochemistry results. Therefore, the authors decided and agreed not to include the results about the frequency of leukocyte cell subsets (old Figure 1 and old Table 1) in the new version  of the manuscript.  If we omit Figure 1 and Table 1, the introduction the aim of the study, the line of discussion and the conclusions remain unchanged. We adapted the manuscript accordingly and removed all data related to immunohistochemistry from the new version of the manuscript (Abstract, Results and Materials and Methods). The parts that we plan to omit in the new version of the manuscript have been crossed out.

 -Figure 6 and table 2 can be provided as supplementary material

Figure 6 (=New Figure 5) and Table 2 (New Table 3) are provided as Supplementary Materials.

Thank you for critical reading the manuscript, for  the valuable suggestions and the time you spent reading the manuscript.

Reviewer 3 Report

Comments and Suggestions for Authors

This is an interesting article describing some pathogenesis of purpura nephritis.

I have several concerns which should be addressed adequately.

1. The rationale of indication of renal biopsy for children with IgA vasculitis in the authors' institution should be added.

2. How many glomeruli examined in each immunostaining? This issue should be added.

3. How influenced therapeutic intervention on the status of macrophage infiltration and the other functional molecules in renal biopsy specimens?

4. How influenced he timing of renal biopsy on the status of macrophage infiltration and the other functional molecules in the renal biopsy specimens?

5. It is nice to add some tables which summarized the results obtained for easily understanding.

Comments on the Quality of English Language

Some minor English editing should be needed.

Author Response

Please find answers to your comments below.

I have several concerns which should be addressed adequately.

  1. The rationale of indication of renal biopsy for children with IgA vasculitis in the authors' institution should be added.

We added in the new version of the manuscript as follows.“ Percutaneous renal biopsy was performed as a routine clinical diagnostic procedure for clinically active IgAVN in patients with 24-hour proteinuria > 1.0 g/dU, persistent 24-hour proteinuria > 0.5 g/dU for at least three months, nephritic and nephrotic syndrome and impaired eGFR [43,44] ….“

  1. How many glomeruli examined in each immunostaining? This issue should be added.

We examine all glomeruli present in the bioptic material and analyzed an average of 7 glomeruli per biopsy material. The range was 5-14 glomeruli. The photomicrographs were taken from the representative samples. We wrote in the new version of the manuscript: „An average of 7 glomeruli (range 5-14 glomeruli) in each sample were visualised, and the representative glomeruli were photographed.“

  1. How influenced therapeutic intervention on the status of macrophage infiltration and the other functional molecules in renal biopsy specimens?

Macrophage activity is not in the guidelines as a parameter to assess the activity of nephritis. Therefore, the authors agree to exclude that part of the Discussion in the new version of the manuscript. Please find it crossed out in the new version of the manuscript with the changes highlighted.

  1. How influenced the timing of renal biopsy on the status of macrophage infiltration and the other functional molecules in the renal biopsy specimens?

We performed labeling about a year ago and the sections labeled using immunohistochemistry method were embedded in Acquatex mounting medium, as it was written in the submitted manuscript. We counted the positive cells then, but we did not take photos at the same time. Aquatex cannot permanently preserve the brown color at the site of specific antigen binding, so the color changed in the most previously positive cells. Therefore we are not able to perform this correlations.

  1. It is nice to add some tables which summarized the results obtained for easily understanding.

We added new Table 2 showing the main results. While answering the questions, we noticed some omissions (lack of arrows) in the Figures and some typos in the text, so we corrected them. References have been aligned with changes in the text and rearranged. Tables and Figures are also renumbered, as the old Figue 1 and old Table 1 are omitted, in the accordance with the suggestions.

Comments on the Quality of English Language

Some minor English editing should be needed.

The manuscript is edited and re-edited in Editage – Professional editing service.

We can deliver the Certificate, if necessary.

We have accepted corrections after the English language editing.

Thank you for the critical reading the manuscript and valuable suggestions.

Round 2

Reviewer 2 Report

Comments and Suggestions for Authors

Issues have been addressed

Reviewer 3 Report

Comments and Suggestions for Authors

I understand the authors' responses. I have no further concerns.